# The Contributions of Cognitive Abilities to the Relationship between ADHD Symptoms and Academic Achievement

**DOI:** 10.3390/brainsci12081075

**Published:** 2022-08-13

**Authors:** Demi Tsantilas, Alzena Ilie, Jessica Waldon, Melissa McGonnell, Penny Corkum

**Affiliations:** 1School Psychology Program, Faculty of Education, Mount Saint Vincent University, Halifax, NS B3M 2J6, Canada; 2Clinical Psychology Program, Department of Psychology and Neuroscience, Dalhousie University, Halifax, NS B3H 4R2, Canada

**Keywords:** ADHD, inattention, hyperactivity, impulsivity, academic achievement, standardized tests, cognitive processes, behavioural symptoms, background risk factors

## Abstract

The main objective of this study was to examine whether increased levels of inattentive (INA) and hyperactive/impulsive (H/I) behaviours were associated with lower scores on standardized tests of achievement in basic reading, spelling, and math skills, after accounting for certain known background risk factors and cognitive processes. Clinical assessment data from a rigorously diagnosed, stimulant-medication-naïve sample of 354 elementary school-aged children experiencing academic difficulties and behavioural symptoms of inattention and/or hyperactivity/impulsivity were analyzed. Although higher scores of INA were significantly associated with lower scores in reading, spelling, and math, these associations did not persist when cognitive variables were added to the models. H/I was associated with math achievement, along with cognitive and background variables. Overall, cognitive variables accounted for the majority of the variance across basic reading, spelling, and math skills. Additionally, the only background demographic variables associated with academic achievement were age and sex for spelling and math. This finding highlights the importance of looking beyond observable INA and H/I behaviours to determine the underlying factors influencing academic achievement. Accurate identification of deficits in specific academic skills and the underlying factors influencing achievement in these skills are essential components in determining appropriate recommendations and targeted interventions.

## 1. Introduction

Academic underachievement is a key indicator of long-term negative functional outcomes across the lifespan. Academically underachieving individuals are more likely to experience employment problems, earn a substantially lower income, depend on government assistance, be involved in criminal activity, use illicit substances, and have poorer health compared to individuals with higher academic achievement [1]. These adverse outcomes in adulthood lead to interpersonal difficulties and familial dysfunction, and place a burden on society through costs associated with government assistance, criminal activity, poor health, and low tax contributions [2,3]. With the multitude of adverse psychosocial outcomes associated with academic underachievement, it is imperative to understand which factors increase the risk of academic underachievement.

Academic underachievement is experienced by up to 80% of students with attention-deficit/hyperactivity disorder (ADHD) [4,5], and is the main reason students with ADHD are referred for clinical assessment [6]. Difficulties with attention and/or hyperactivity/impulsivity are thought to manifest in the form of behaviours that interfere with the development of the fundamental skills essential to learning [7]. Behavioural symptoms of inattention (INA) and hyperactivity/impulsivity (H/I) are also evident in other disorders, such as Learning Disabilities (LD) [8], Anxiety Disorders [9,10], Major Depressive Disorder (MDD) [11,12], Obsessive Compulsive Disorder (OCD) [13], Autism Spectrum Disorder (ASD) [14], Obstructive Sleep Disorder (OSD) and other sleep disorders [15], and exposure to trauma or Post-Traumatic Stress Disorder (PTSD) [16]. Students experiencing high levels of INA and/or H/I without meeting diagnostic criteria for ADHD experience academic difficulties similar to those of students with formal diagnoses of ADHD [6,17,18]. The diagnostic framework used to define and diagnose ADHD is in the Diagnostic and Statistical Manual of Mental Disorders, Fifth Edition (DSM-5) [19]. Given the large number of students with behavioural symptoms of INA and H/I, it is important to understand the association between these behaviours and academic achievement at the symptom level rather than at the diagnosis level.

Previous studies have demonstrated a negative linear relationship between INA symptom severity (i.e., the number of symptoms present) and academic achievement [20,21]. INA has been found to predict literacy and math achievement [22] and has been associated with impairments in the acquisition of reading skills, such as decoding, fluency, comprehension, and vocabulary [23,24]. Additionally, INA has been found to predict spelling accuracy, handwriting fluency [25], and fine-motor control [26]. Higher levels of INA in kindergarten students have been shown to predict long-term reading impairments [27]. Furthermore, INA has been found to be a stronger predictor of academic difficulties than the presence of an LD [28]. Higher levels of teacher-rated INA behaviour have consistently been associated with lower scores on standardized tests of academic achievement and classroom performance outcomes [29].

Symptoms of H/I are more prominent in younger children and tend to decrease with age [19]. Behavioural symptoms of H/I have weaker associations with academic impairments than INA behaviours, but are closely related to disruptive classroom behaviour, lower grades, grade retention, suspension, and expulsion [28]. H/I symptoms are stronger predictors of these outcomes than physical health problems, with the associations persisting even when scores of H/I symptoms are below the diagnostic threshold for ADHD [30].

Academic underachievement has also been associated with deficits in cognitive abilities. Specifically, working memory, processing speed, and visuospatial ability have been identified as cognitive abilities that account for a large proportion of the variance in academic achievement [31]. Working memory has been associated with reading ability [32,33,34], reading comprehension [32], spelling [35], writing ability [36,37], and math problem-solving ability [33,34,38,39]. Slow processing speed has been shown to predict reading difficulties in decoding, fluency, and comprehension [40], as well as math difficulties [33]. Deficits in visual-spatial abilities have been associated with difficulties in mathematics [41], reading acquisition [42], and spelling [43]. Additionally, visual-motor integration abilities have been found to predict spelling and writing composition scores [44], and general reading and math achievement [45].

The existing literature has identified several demographic factors that are associated with academic achievement, regardless of behaviours related to symptoms of INA and/or H/I, learning and other mental health disorders, and cognitive abilities. Increasing age [46], being male [47], low socioeconomic status (SES) [48,49], alternative family structures such as single-parent families [50], and low birthweight [51] have all been associated with academic underachievement. Therefore, studies investigating predictors of academic achievement in children experiencing difficulties with INA and H/I should include demographic factors known to influence academic achievement [46].

Despite the abundance of literature demonstrating associations between academic underachievement and background demographic factors, cognitive factors, and behavioural symptoms of INA and H/I, very few studies have included demographic, cognitive, and behavioural predictors to determine their unique contributions to academic achievement [30]. Most studies investigating the associations between academic achievement and INA and H/I at the symptom level have been conducted with community samples of children whose symptom counts are measured based on rating scales of parent- or teacher-rated behavioural symptoms, without having undergone rigorous assessment procedures [7,52]. The absence of multi-informant rigorous assessment procedures is problematic, as parents and teachers observe children in different contexts and tend to rate INA and H/I behaviours differently [53], which is demonstrated by the low (*r* < 0.30; [53]) to moderate (*r* = 0.40–0.50; [54]) interrater agreement typically observed on rating scales.

Additionally, most studies investigating academic achievement in students experiencing difficulties with INA and/or H/I have methodological flaws. These studies often use an operational definition of ADHD that does not require rigorous diagnostic procedures [7,52] As such, they often include children who do not have a formal diagnosis of ADHD and exclude children who have insufficient INA and H/I symptoms to meet the operational definition [7]. This limits our understanding of the association between academic achievement and symptoms of INA and H/I at the symptom level as a continuously distributed trait. Studies also often include children who are receiving medication for the treatment of ADHD. Stimulant medications have been shown to improve INA and H/I behaviours to varying degrees [55], which could potentially confound results [46,56].

### The Current Study 

The objective of the present study was to examine the associations between academic achievement (i.e., basic reading, spelling, and math skills) and behavioural symptoms of INA and H/I, after accounting for known background risk factors (i.e., age, sex, SES, family structure, and low birthweight) and cognitive processes (i.e., working memory deficits, slower processing speed, impairments in visual-spatial abilities), in a rigorously diagnosed, stimulant-medication-naïve, clinical sample of elementary school-aged children. This study’s research question was to determine the association between INA and H/I symptom severity and academic achievement, and if it continued to exist after accounting for background risk factors and cognitive processing impairments. To answer this research question, clinical assessment data from a rigorously diagnosed sample of 354 elementary school-aged children experiencing academic difficulties and behavioural symptoms of INA and/or H/I were analyzed. Based on current existing literature, it was hypothesized that higher scores of INA, not H/I, after accounting for background risk factors and cognitive processing variables, would continue to be associated with lower scores in basic reading, spelling, and math skills in elementary school-aged children.

## 2. Materials and Methods

### 2.1. Participants

Participants in this study consisted of children experiencing difficulties with attention and/or hyperactivity/impulsivity who were referred to and assessed at the Colchester East Hants ADHD Clinic from 2000 up to June 2019. Children eligible to be assessed at the clinic were between 6 and 12 years of age at the time of their assessment; resided within the Colchester East Hants health catchment area; had not previously received an ADHD diagnosis; had never been on medication for inattention, hyperactivity, impulsivity, or any other mental health disorder; and had not received a psychoeducational assessment in the two years prior to their clinical assessment. Only children whose parents consented at the time of the assessment to the inclusion of their children’s assessment/diagnostic data in potential future research were included in this study.

### 2.2. Measures

#### 2.2.1. Background Information Form

The Background Information Form is a standardized parent self-report questionnaire based on the Family and Household Form from the Ontario Child Health Study [57]. The Background Information Form was completed by all parents of children assessed by the ADHD Clinic. Background risk factors that were used in this study as predictors of academic achievement were drawn from this measure and include sex (categorical dichotomous variable coded as 0 = Female, 1 = Male); age (measured in months); family structure (categorical dichotomous variable coded as 0 = Two-Parent Household, 1 = Single-Parent Household); birthweight (measured in ounces); and SES (measured by the highest level of education of the parent with the highest level of education; categorical variable coded on a scale of 1 to 8 [1 = some elementary; 2 = completed elementary; 3 = some secondary; 4 = completed secondary; 5 = some community or technical college; 6 = completed community or technical college; 7 = some university or teachers college; 8 = completed university or teachers college]). Parent educational attainment was used as a univariate proxy measure of SES, as education has been found to be the strongest predictor of occupational status and income [58,59].

#### 2.2.2. Diagnostic Interviews

Two semi-structured diagnostic interviews were conducted as part of the clinical assessment.

Parent Interview for Child Symptoms (PICS) [60]. The PICS is a semi-structured three-module diagnostic interview designed to systematically assess symptoms of ADHD and other childhood disorders based on DSM-IV/DSM-5 diagnostic criteria. The PICS was administered by a clinical psychologist and a pediatrician, who scored parent responses describing the child’s behaviour in a variety of situations. The severity of each of the nine INA and nine H/I symptoms was rated on a 3-point scale: 0 = absent; 1 = trivial abnormality; 2 = definite abnormality, with symptoms rated as 2 regarded as clinically significant. The PICS has good reliability for diagnosing mental health disorders in children [61].

Teacher Telephone Interview (TTI) [62]. The TTI is a semi-structured diagnostic interview designed to systematically assess behaviour and functioning in a school setting and assess symptoms of ADHD and other childhood mental health disorders based on DSM-IV/DSM-5 diagnostic criteria. The TTI, which is designed to be used in conjunction with the PICS, was administered by a clinical psychologist and pediatrician in a telephone interview with the child’s teacher prior to the clinic day. Teacher responses describing the child’s behaviour in a school setting are scored on the same scale as the PICS (see above). The TTI-IV has good interscorer (r = 0.96–0.98) and test–retest reliability (r = 0.79–0.95) for ADHD symptoms [63].

The PICS and the TTI were used for diagnostic purposes and provided INA and H/I symptom counts for analyses. A symptom was considered present if it was rated as a 2, using the above described 3-point scale. Total PICS and TTI symptom counts for INA and H/I symptoms were combined and used as behavioural predictors of academic achievement.

#### 2.2.3. Wechsler Intelligence Scale for Children (WISC)

The WISC is an individually administered standardized measure of cognitive ability designed to provide a comprehensive assessment of general intellectual functioning in children from ages 6 to 16 years. In the current study, four index scores were used as measures of specific cognitive abilities: The Verbal Comprehension Index (VCI), Visual Spatial Index (VSI), Working Memory Index (WMI), and Processing Speed Index (PSI).

Three versions of the WISC (Third Edition (WISC-III), [64]; Fourth Edition (WISC-IV), [65]; and Fifth Edition (WISC-V), [66]) have been released in the 20 years that the ADHD Clinic has been in operation, and all have been used by the Clinic, as measures were updated to reflect current best practices. Strong correlations between WISC-III and WISC-IV index scores (0.72–0.88) [65] and WISC-IV and WISC-V index scores (0.58–0.85) [66] supported using data obtained from these three versions as measures of cognitive abilities in the current study.

Additionally, the most recent version of the WISC separated the Perceptual Reasoning Index (PRI) from the previous version (WISC-IV) into the Visual Spatial Index (VSI) and the Fluid Reasoning Index (FRI). Given that the PRI and the VSI are conceptually similar (i.e., measure the ability to evaluate visual details and to understand visual–spatial relationships to construct designs from a model) [66], and given the higher reported correlation [66] between the PRI and VSI (0.73) compared to the correlation between PRI and FRI (0.58), the VSI (WISC-V) and PRI (WISC-IV) were included as measures of the same cognitive ability. The FRI from the WISC-V was not included in this study.

#### 2.2.4. Beery–Buktenica Developmental Test of Visual–Motor Integration (VMI)

The VMI is a standardized measure designed to assess the ability to effectively coordinate vision, perception, and fine motor movements to execute hand/finger tasks. The VMI can be administered to children and adults from ages 2 to 100 years. Only the standard score for the primary subtest, in which the task requires the individual to copy designated geometric shapes of increasing complexity, was used in this study. The VMI has demonstrated good inter-rater and test-retest reliability, and good concurrent and construct validity [67]. Although three editions of the measure have been used by the clinic in the 20 years it has been in operation, all editions of the VMI contain the same items, and the reported correlations of the VMI-6 with the two previous editions are high (0.98 and 0.99, respectively) [67]. Therefore, standard scores from the VMI-4, VMI-5, and VMI-6 were used in the current study as a measure of visual-motor integration.

The four WISC indices (VCI, VSI, WMI, PSI) and the VMI index, all of which are measured in standard scores (*M* = 100, *SD* = 15), were used as cognitive predictors of academic achievement.

#### 2.2.5. Wechsler Individual Achievement Test (WIAT)

The WIAT is an individually administered academic achievement test designed to assess listening, speaking, reading, writing, spelling, and mathematics skills in individuals 4 to 50 years old. Standard scores from three WIAT subtests were used as indicators of academic achievement: Word Reading, Spelling, and Numerical Operations. WIAT scores are represented as standard scores (*M* = 100, *SD* = 15).

The WIAT–III is a revision of the WIAT-Second Edition (WIAT–II; [68]), which has also been used in assessments at the ADHD Clinic. The WIAT-II and WIAT-III have been found to have high internal consistency and test–retest reliability [68,69]. Each version of the WIAT is co-normed with and empirically linked to the version of the WISC in use during the same period (i.e., WIAT-II with WISC-III and WISC-IV), allowing for valid and reliable comparison between achievement and ability [64,66,69].

### 2.3. Procedure

Full ethical approval for this study was granted through the Nova Scotia Health Authority and Mount Saint Vincent University Research Ethics Boards.

The current study is an archival data analysis study. Data for this study were drawn from the ADHD Clinic’s clinical/research database, which included assessment and diagnostic data of children assessed by the clinic between 2000 and 2019. Assessments completed at the clinic were comprehensive and rigorous, using recommended evidence-based assessment guidelines and an interdisciplinary diagnostic approach. In brief, prior to arriving at the ADHD Clinic, participants’ parent(s) and teacher(s) completed screening measures [70], and a demographic questionnaire. Each assessment included a review of the child’s school records and a classroom observation, semi-structured diagnostic interviews with the PICS and TTI, and a standardized psycho-educational assessment battery with the child (WISC, VMI, WIAT). A detailed description of this clinic can be found in the article by McGonnell and colleagues [71].

### 2.4. Statistical Analyses

Data for the current study were analyzed using MS Excel (version 16.38; Microsoft Corporation, United States) and IBM SPSS software (version 26;Armonk, NY: IBM Corp). Data were drawn from the ADHD Clinic’s clinical/research database. Descriptive statistics were first computed to provide an overview of the sample characteristics. Prior to conducting analyses, the pattern of missing data was examined using Little’s Missing Completely at Random (MCAR) test [72,73], which indicated that the data were missing at random, *χ*^2^(91) = 94.710, *p* = 0.37. Cases with missing values (*n* = 90) were excluded from the study. Additionally, all assumptions for hierarchical multiple linear regressions were met.

**Hypothesis:** Higher scores of INA, not H/I, will continue to be associated with lower scores in basic reading, spelling, and math skills in elementary school-aged children, after accounting for background risk factors (age, sex, SES, family structure, and low birthweight), and cognitive processing variables (WISC VCI, VSI, WMI, PSI, and VMI score).

Three hierarchical multiple linear regressions were conducted to evaluate the contribution of INA and H/I to reading, spelling, and math, after accounting for background risk factors and cognitive variables. Three WIAT subtests were used as dependent variables in each of the regressions. The same independent variables were entered in each step of each of the three regressions. Background risk-factors were entered in the first step, including sex, age, SES, family structure, and birthweight. Cognitive variables were entered in the second step, including the four WISC index scores (VCI, VSI, WMI, PSI) and the VMI index score. Behavioural symptoms were entered in the third step of the regressions, including the total combined INA and H/I symptom counts across the PICS and TTI.

## 3. Results

### 3.1. Sample Characteristics

The final sample consisted of 354 participants between the ages of 6 and 12 years (*M* = 8.56, *SD* = 1.62). The majority of participants (*n* = 247, 70%) were male. Participants’ birthweight ranged from 2 lbs. 4 oz. to 11 lbs. 1 oz. (*M* = 7.55, *SD* = 1.41), as reported by the parent. Twenty-eight (0.6%) participants were born at a weight that is considered low birthweight (i.e., below 2500 g, or 5 lbs. 8 oz.). Participant grade level ranged from Primary/Kindergarten to Grade 7 (*M* = 2.82, *SD* = 1.62). Based on the parent report, the majority of participants (*n* = 282, 80%) were from two-parent families, and on average, parents had completed community or technical college (*M* = 5.95, SD = 1.63). Demographic characteristics are summarized in Table 1.

Approximately 10% (*n* = 37) of the sample did not reach the diagnostic criteria for ADHD, LD, or a Mental Health (MH) disorder. Just over half (52%, *n* = 184) of participants were diagnosed with ADHD (Subtype: INA (*n* = 56), H/I (n=15), Combined: *n* = 113). Approximately one-fifth had ADHD only (*n* = 74), while the others had comorbid LD (*n* = 70), MH disorders (*n* = 22), or both MH and LD (*n* = 18). The second most frequent diagnosis was LD (28%, *n* = 98), either as the only diagnosis (*n* = 70) or along with another MH diagnosis (not ADHD) (*n* = 28). The remaining children (10%, *n* = 35) were diagnosed with one or more MH disorders (not ADHD or LD, e.g., Generalized Anxiety Disorder, Autism Spectrum Disorder). The clinical description of the sample is summarized in Table 2.

### 3.2. Sample Clinical Description

#### 3.2.1. Behavioural Symptoms (INA and H/I)

Combined parent (PICS) and teacher ratings (TTI) of INA and H/I symptoms indicated a mean symptom count for INA of 10.23 (*SD* = 4.44) and for H/I of 8.10 (*SD* = 5.20). PICS-rated symptoms of INA had a mean symptom count of 4.72 (*SD* = 2.86), and TTI-rated symptoms of INA had a mean symptom count of 5.52 (*SD* = 2.72). PICS-rated symptoms of H/I indicated a mean symptom count of 4.41 (*SD* = 3.04), and TTI-rated symptoms of H/I indicated a mean symptom count of 3.69 (*SD* = 3.14).

#### 3.2.2. Cognitive Ability

Participants’ mean standard scores on WISC indices all were in the Average range: PSI (*M* = 90.96, *SD* = 13.36), WMI/FDI (91.32, *SD* = 12.75), POI/VSI/PRI (100.17, *SD* = 13.08), and VSI (*M* = 100.17, *SD* = 14.08). The VMI Mean score was 92.16 (*SD* = 12.24).

#### 3.2.3. Academic Achievement

Participants’ mean standard scores on WIAT subtests were in the Low Average to Average range: Numerical Operations (*M* = 88.06, *SD* = 13.54), Spelling (*M* = 89.91, *SD* = 13.83), and Word Reading (*M* = 93.63, *SD* = 15.92). Overall, approximately one-third of participants were underachieving (one or more *SD* below the mean): Word Reading: 31% (*n* = 108); Spelling: 34% (*n* = 120), and Numerical Operations 38% (*n* = 136).

*Reading.* Model 1 (background risk-factors) was statistically significant, F(5, 348) = 2.28, *p* = 0.047, R^2^ = 0.03. Sex was the only significant predictor of reading, β = 0.12, t(348) = 2.15, *p* = 0.03. Model 2 (cognitive variables) was statistically significant, ΔF(5, 343) = 38.01, *p* < 0.001, ΔR^2^ = 0.35. Significant predictors in Model 2 were: VCI, β = 0.36, t(343) = 6.57, *p* < 0.001; WMI, β = 0.29, t(343) = 5.62, *p* < 0.001; and VMI, β = 0.12, t(343) = 2.40, *p* = 0.02. Model 3 (behavioural symptoms) accounted for 0.7% of the variance and was not statistically significant, ΔF(2, 341) = 2.06, *p* = 0.13, ΔR^2^ = 0.007. The results are summarized in Table 3.

*Spelling.* Model 1 (background risk-factors) was not statistically significant, F(5, 348) = 1.47, *p* > 0.20, R^2^ = 0.02. Model 2 (cognitive variables) was statistically significant, ΔF(5, 343) = 27.10, *p* < 0.001, ΔR^2^ = 0.28. The significant predictors in Model 2 were: VCI, β = 0.24, t(343) = 4.20, *p* < 0.001; WMI, β = 0.31, t(343) = 5.66, *p* < 0.001; and VMI, [β = 0.17, t(343) = 3.14, *p* = 0.002]. Age, which was statistically significant in Model 1, remained significant in Model 2. SES (measured by highest level of parent education) was a significant predictor of spelling scores, β = −0.15, t(343) = −3.11, *p* = 0.002. Model 3 (behavioural symptoms) was not statistically significant, ΔF(2, 341) = 0.74, *p* = 0.48, ΔR^2^ = 0.003. The results are summarized in Table 3.

*Math.* Model 1 (background risk-factors) was statistically significant, F(5, 348) = 6.86, *p* > 0.001, R^2^ = 0.09. The significant predictors in Model 1 were sex, β = 0.12, t(348) = 2.38, *p* = 0.02; age, β = −0.21, t(348) = −4.14, *p* < 0.001; and SES, β = 0.13, t(348) = 2.55, *p* = 0.01. Model 2 (cognitive variables) was statistically significant, ΔF (5, 343) = 34.65, *p* < 0.001, ΔR^2^ = 0.31. All cognitive variables entered in Model 2 were significant predictors of math scores: VCI: β = 0.20, t(343) = 3.59, *p* < 0.001; VSI: β = 0.14, t(343) = 2.41, *p* = 0.02; WMI: β = 0.21, t(343) = 4.12, *p* < 0.001; PSI: β = 0.13, t(343) = 2.69, *p* = 0.007; and VMI: β = 0.12, t(343) = 2.31, *p* = 0.02. Sex and age, which were statistically significant predictors in Model 1, remained significant in Model 2. Model 3 (behavioural symptoms) was statistically significant, ΔF (2, 341) = 4.24, *p* = 0.02, ΔR^2^ = 0.02. H/I significantly predicted math scores, β = −0.13, t(341) = −2.60, *p* = 0.01. Cognitive scores entered in Model 2 remained significant in Model 3. Additionally, sex and age remained statistically significant predictors in Model 3. The results are summarized in Table 3.

## 4. Discussion

The goal of the present study was to determine whether increased levels of behavioural symptoms of INA and H/I were associated with lower scores on standardized tests of achievement in basic reading, spelling, and math skills, after accounting for known background risk factors and cognitive processes. Based on the background literature, it was expected that higher scores of INA, not H/I, would be associated with lower scores in basic reading, spelling, and math skills after accounting for background risk factors and cognitive processing variables. This hypothesis was not supported. Although higher scores of INA were significantly associated with lower scores in reading, spelling, and math initially, these associations did not persist when cognitive variables were added to the models. Overall, cognitive variables accounted for the majority of the variance across basic reading, spelling, and math skills. VCI, WMI, and VMI were statistically significant predictors for reading and spelling, and all cognitive variables (VCI, VSI, WMI, PSI, VMI) were statistically significant predictors for math.

Previous literature suggests that subjectively measured INA behaviours alone cannot solely account for the academic underachievement experienced by students [74,75]. INA behaviour reduction through treatment with stimulant medication produces questionable improvements in academic achievement, particularly on standardized measures [4,6,76,77]. Empirical research has suggested that academic underachievement experienced by children with high levels of INA is a product of deficits in cognitive abilities, as INA is thought to be characterized by the behavioural manifestation of underlying impairments in cognitive abilities [74,78,79].

Given the associations between high levels of INA, academic underachievement, and deficits in cognitive abilities, it is likely that deficits in cognitive abilities associated with INA are driving the relationship between high levels of INA and academic underachievement [78,79]. This conceptualization of the associations between INA, cognitive deficits, and academic underachievement is consistent with the results of the present study, in which cognitive variables accounted for the largest proportion of the variance in academic achievement. Another interpretation of this finding could be that cognitive variables are stronger predictors of academic achievement than INA at the developmental stage of the present sample. The majority of the sample consisted of children in lower elementary grades, and INA is thought to exert a greater effect on academic achievement as academic demands increase in higher grades [7,80]. The age of the participants could also potentially explain the relatively low proportion of the variance that INA accounted for initially, before including the demographic and cognitive variables in the models.

Higher levels of H/I behaviours were not expected to be associated with lower scores on basic reading, spelling, and math skills, as previous research suggests that H/I is an inconsistent predictor of academic achievement [28,29,46]. As expected, higher levels of H/I did not significantly predict reading and spelling scores. However, results indicated that higher levels of H/I significantly predicted lower scores in math. Several studies have demonstrated associations, albeit inconsistently, between increasing levels of H/I and decreased math scores [30,81]. Furthermore, students with high levels of H/I behaviours tend to make careless mistakes on math tasks, which could potentially explain this association.

Previous research has stressed the importance of including background risk factors as covariates in research investigating INA and academic achievement to avoid potential confounds [46]. In the present study, background demographic variables were not consistently associated with academic achievement across academic domains. No background variable was significantly associated with reading scores after cognitive variables were added to the model. For spelling, age and SES were significant predictors when cognitive variables were added to the model. However, spelling scores decreased as SES increased, which was unexpected and counterintuitive. No studies were found in the literature to explain this association. Furthermore, although math scores were initially associated with age, sex, and SES, only age and sex continued to be associated with math scores when cognitive and behavioural symptoms were added to the model. Overall, background demographic variables identified by previous research as risk factors for academic achievement were inconsistently associated with reading, spelling, and math scores in the current study.

Approximately one-third of the children in this sample were underachieving in specific academic domains, with the greatest proportion underachieving in math, and the lowest proportion underachieving in reading. Given that the sample in this study consisted of children experiencing difficulties with ADHD symptoms and learning, it would be expected that this proportion would be higher, as studies have reported that as many as 80% of students with these difficulties experience academic impairments [4,5]. This finding highlights the importance of standardized testing, as teacher-rated underachievement based on subjective measures does not necessarily indicate deficits in academic skills [82]. Although subjective measures are important indicators of future academic success and educational attainment [83], they are often unable to accurately quantify academic skills [82]. Therefore, when deficits in academic skills are suspected, standardized achievement tests should be included in comprehensive assessments to determine the presence or absence of academic impairments.

### 4.1. Strengths, Limitations, and Future Directions

This study included a large, rigorously diagnosed, stimulant-medication-naïve clinical sample. Very few studies investigating the association between INA and H/I behaviours and academic achievement include rigorously diagnosed samples, and even fewer studies include rigorously diagnosed samples of this size. The large sample size (N = 354) increased the generalizability of the results. Additionally, the comprehensive, evidence-based assessment methods used at the Clinic are considered the gold standard in the assessment and diagnosis of ADHD [55], and their use increases the accuracy and validity of the INA and H/I symptom counts included in this study. Previous research has highlighted the importance of including risk factors known to influence academic achievement, as well as INA and H/I behaviours, as covariates in studies investigating the associations between these behaviours and academic achievement [46]. The present study included known risk factors and accounted for their unique contribution to academic achievement. The use of standardized achievement measures provides accurate quantifications of deficits in academic skills and holds clinical utility, as they are frequently used to determine diagnoses in clinical and educational settings.

Certain limitations to the present study should be acknowledged. Although the dimensional measurement of INA and H/I behaviours allowed for the inclusion of all participants assessed by the clinic, the sample in this study was a clinically referred sample of children who were displaying some symptoms of INA and/or H/I. Therefore, the INA and H/I symptom counts in this study were elevated compared to INA and H/I behaviours in the general population. Future research should consider including non-referred children in addition to clinically referred children to investigate these associations. Additionally, the majority of the present sample consisted of children in lower elementary grades with the mode being Grade 2. Given the young age of the participants in this study and the developmental course of INA and H/I symptoms [19], it is possible that the results in this study would not generalize to an older sample. Future research could consider investigating these results with older children. Although the present study included a number of background risk factors known to influence the relationship between INA and academic achievement, certain important risk factors that affect academic achievement were not included, such as childhood adversity, physical health conditions, and other cognitive variables (e.g., overall memory abilities, phonological processing skills). Future research could examine these associations including these known predictors of academic achievement as covariates. Additionally, the comorbidity of ADHD and LD in this sample was higher than average (47.8%), and the comorbidity of ADHD and MH was lower than average (21.7%), which limits the generalizability of these findings to the general population [84,85]. This may be in part due to the younger age of this sample, and the fact the clinic was a partnership between mental health and the school board. This means that there was likely a referral bias toward children with learning problems rather than mental health problems.

### 4.2. Clinical and Educational Implications

Results from the current study indicated that although INA significantly predicted academic achievement, the association did not persist after accounting for cognitive abilities. This finding emphasizes the importance of administering psycho-educational batteries and including measures of cognitive variables in the assessment of students demonstrating high levels of INA and/or H/I behaviours and academic difficulties. Additionally, the present study underscores the importance of understanding INA and H/I behaviours at the symptom level rather than the diagnosis level. Although all children in the sample were thought to possibly have ADHD, only half the sample met diagnostic criteria. Although INA and H/I behaviours are core symptoms of ADHD, they are also characteristic of other conditions or MH disorders, which emphasizes the importance of identifying the factors underlying INA or H/I behaviour through differential diagnosis procedures.

## 5. Conclusions

The present study indicated that, while increasing levels of INA behaviours were associated with academic underachievement in basic reading, spelling, and math skills, these associations did not persist when cognitive abilities were taken into account. This finding highlights the importance of looking beyond observable INA and H/I behaviours to determine the underlying cognitive variables influencing academic achievement. Importantly, the accurate identification of deficits in specific academic skills, and the underlying cognitive processing factors influencing achievement in these skills are essential components in determining appropriate recommendations and targeted interventions. Targeted interventions could, in turn, lead to greater academic success in students with academic difficulties. Given the well-established association between academic underachievement and decreased educational, psychosocial, and functional outcomes across the lifespan, improving a student’s academic functioning can potentially lead to subsequently improved life outcomes.

## Figures and Tables

**Table 1 brainsci-12-01075-t001:** Demographic characteristics of total sample (*N* = 354).

Demographic Characteristics	*N* (%)	*M (SD)*	Minimum	Maximum
Sex				
Male	247 (70%)	-	-	-
Female	107 (30%)	-	-	-
Age	-	8.56 (1.62)	5.42	12.58
Grade	-	2.80 (1.65)	0	7
Parental Education	-	5.95 (1.63)	3	8
Family Structure				
1-Parent Family	72 (20%)	-	-	-
2-Parent Family	282 (80%)	-	-	-
Birthweight	-	7.55 (1.41)	2.25	11.06

*Note.* Age measured in years. Grade measured in numeric grade level (0 = Grade Primary). Parental Education was used as a univariate proxy measure of SES, and was measured by the highest level of education attained by the parent with the highest education on a scale of 1 to 8 (1 = some elementary; 2 = completed elementary; 3 = some secondary; 4 = completed secondary; 5 = some community or technical college; 6 = completed community or technical college; 7 = some university or teachers college; 8 = completed university or teachers college). Birthweight measured in pounds.

**Table 2 brainsci-12-01075-t002:** Clinical description of sample indicating number and percentages of mental health disorders diagnosed in sample.

Mental Health Diagnosis	Total Sample *N =* 354	
	*n*	%
ADHD Total	184	52
ADHD Presentation		
ADHD-PI	56	16
ADHD-HI	15	4
ADHD-C	113	32
Learning Disability (LD)	186	53
MH Dx other than ADHD and/or LD	103	29
Oppositional Defiant Disorder	21	6
Generalized Anxiety Disorder	20	6
Autism Spectrum Disorder	16	5
Sleep Disorders	15	4
Specific Phobia	10	3
Enuresis	9	3
Tourette’s Disorder	9	3
Chronic Motor Tics	6	2
Stereotypical Movements	5	1
Conduct Disorder	4	1
Obsessions	4	1
Other MH Disorders ^a^	19	5
No Diagnosis	37	11

*Note.* ADHD = Attention-Deficit/Hyperactivity Disorder. ADHD-PI = Attention-Deficit/Hyperactivity Disorder Predominantly Inattentive Presentation. ADHD-HI = Attention-Deficit/Hyperactivity Disorder Hyperactive/Impulsive Presentation. ADHD-C = Attention-Deficit/Hyperactivity Disorder Combined Presentation. LD = Learning Disability. LD includes students considered at-risk for Learning Disability. MH Dx = Mental Health Diagnosis/es. ^a^. Other MH Disorders = Disorders that were diagnosed in less than 1% (3 or fewer counts per diagnosis) of the total sample and include diagnoses of Acute Traumatic Stress, Body Dysmorphic Disorder, Chronic Vocal Tics, Compulsions, Dysthymic Disorder, Encopresis, Major Depressive Disorder, Mania/Hypomania, Separation Anxiety Disorder, Social Phobia, Transient Tic Disorder.

**Table 3 brainsci-12-01075-t003:** Hierarchical multiple linear regression analyses predicting academic achievement in reading (WIAT Word Reading Subtest), spelling (WIAT Spelling Subtest), and mathematics (WIAT Numerical Operations Subtest) from background risk factors, cognitive variables, and total symptom count of parent- and teacher-rated inattention and hyperactivity/impulsivity.

	Domain of Academic Achievement					
	Reading		Spelling		Mathematics	
Predictor	Δ*R*^2^	β	Δ*R*^2^	β	Δ*R*^2^	β
Step 1 Background Risk Factors	0.03 *		0.02		0.09 ***	
Sex		−0.12 *		−0.04		−0.12 *
Age		−0.06		−0.14 *		−0.21 ***
Birthweight		−0.06		−0.03		−0.03
SES		−0.07		−0.03		−0.13 *
Family Structure		−0.03		−0.00		−0.05
Step 2 Cognitive Variables	0.35 ***		0.28 ***		0.31 ***	
VCI		−0.36 ***		−0.24 ***		−0.20 ***
VSI		−0.07		−0.03		−0.14 *
WMI		−0.29 ***		−0.31 ***		−0.21 ***
PSI		−0.09		−0.03		−0.13 **
VMI		−0.12 *		−0.17 **		−0.12 *
Step 3 Behavioural Symptoms	0.007		0.003		0.02 *	
INA−Total		−0.04		−0.02		−0.002
HI−Total		−0.11 *		−0.07		−0.13 **
Total *R*^2^	0.38 ***		0.30 ***		0.41 ***	
*n*	354		354		354	

* *p* < 0.05 ** *p* < 0.01 *** *p* < 0.001. *Note:* VCI = Verbal Comprehension Index; VSI = Visual Spatial Index; WMI = Working Memory Index; PSI = Processing Speed Index; VMI = Visual-Motor Integration.

## Data Availability

Not applicable.

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
