# Peer review of "The Contributions of Cognitive Abilities to the Relationship between ADHD Symptoms and Academic Achievement"

_brainsci, 2022, doi:10.3390/brainsci12081075_

Round 1
Reviewer 1 Report
In the introduction, the authors describe the current state of the problem of ADHD. Although the difficulties of such children are described, the authors do not provide an overview of the different approaches to defining and studying ADHD. To date, there is no consensus on the causes and factors of ADHD, and the authors should describe this. It is interesting to consider that children with ADHD and mild cognitive impairment differ in their cognitive manifestations.
I suggest studying the papers:
https://jamanetwork.com/journals/jamanetworkopen/fullarticle/2698633
https://link.springer.com/chapter/10.1007/978-3-030-71637-0_6
https://link.springer.com/article/10.1007/s10484-019-09439-x
https://tp.amegroups.com/article/view/30808/28327
https://onlinelibrary.wiley.com/doi/10.1002/brb3.330
https://publications.aap.org/pediatrics/article-abstract/135/4/e994/77059/Prevalence-of-Attention-Deficit-Hyperactivity?redirectedFrom=fulltext
Consequently, the introduction needs to be expanded.
In the second part of the introduction, the authors describe the shortcomings of existing studies, but do not provide references. This needs to be specified.
It is better to move the description of the research sample to the section with materials and methods, rather than leaving it in the results.
The results of the study are described fully and competently. The discussion could be supplemented by choosing an ADHD model that is consistent with the results of this experiment.
The authors describe in detail the limitations of the study, its strengths, and the potential for future use of the results.
Unfortunately, the article did not follow a template, and there were no line numbers, which made the review process difficult.
Author Response
Please see the attachment.
Please see below for our responses. We have put our responses in red.
Reviewer #1 Comments:
In the introduction, the authors describe the current state of the problem of ADHD. Although the difficulties of such children are described, the authors do not provide an overview of the different approaches to defining and studying ADHD. To date, there is no consensus on the causes and factors of ADHD, and the authors should describe this. It is interesting to consider that children with ADHD and mild cognitive impairment differ in their cognitive manifestations.
I suggest studying the papers:
- https://jamanetwork.com/journals/jamanetworkopen/fullarticle/2698633
- https://link.springer.com/chapter/10.1007/978-3-030-71637-0_6
- https://link.springer.com/article/10.1007/s10484-019-09439-x
- https://tp.amegroups.com/article/view/30808/28327
- https://onlinelibrary.wiley.com/doi/10.1002/brb3.330
- https://publications.aap.org/pediatrics/article-abstract/135/4/e994/77059/Prevalence-of-Attention-Deficit-Hyperactivity?redirectedFrom=fulltext
Consequently, the introduction needs to be expanded.
- Thank you for your comments, and we agree there is interesting and important discussions to be had about this topic! However, we did not add more about this topic to the Introduction because the diagnostic approach and etiology of ADHD is outside the scope of this study. Rather, the Introduction is focused on developing the rationale of the study as related to attention and academic performance. We did however add one sentence into the Introduction to ensure the reader understood the diagnostic framework for this study: “The diagnostic framework used to define and diagnose ADHD is in the Diagnostic and Statistical Manual of Mental Disorders, Fifth Edition (DSM-5).”
In the second part of the introduction, the authors describe the shortcomings of existing studies, but do not provide references. This needs to be specified.
- The references have been added to the second half of the introduction (Paragraphs 7 and 8).
It is better to move the description of the research sample to the section with materials and methods, rather than leaving it in the results.
- We referred to other papers published in the Brain Sciences Journal, and most studies discuss the research sample in the first paragraph of the Results section. We added two tables of the demographic characteristics and clinical description of the total sample to the results as well.
The results of the study are described fully and competently. The discussion could be supplemented by choosing an ADHD model that is consistent with the results of this experiment.
- It is unclear what type of model would be appropriate (diagnostic model, theoretical model)? Given that the study was focused on understanding the relationships between attention and academic achievement, it did not seem appropriate to include a model in the Discussion, but rather more appropriate to describe the findings and place these in the context of the extant literature.
The authors describe in detail the limitations of the study, its strengths, and the potential for future use of the results.
- We are glad you found these sections to be appropriately detailed.
Unfortunately, the article did not follow a template, and there were no line numbers, which made the review process difficult.
- We apologize that line numbers were not included. In the future, we will make sure line numbers are included in the original submission.

Reviewer 2 Report
This is excellent and relevant research. The authors have rigorously assessed and diagnosed children referred to their clinic; and all the data collected has allowed for an elegant statistical analysis. I note that the introduction is extremely well written - building the logic of the rationale for this research. The findings are very relevant to the field, as academic underachievement is the main reason for referral for ADHD assessment. The fact that not all of their subjects were diagnosed with ADHD allowed for a dimensional review of inattentive (INA) and hyperactive/impulsive (HI) behaviours, which is very helpful and relevant as well.
Firstly, some smaller points:
1) in the title, ADHD should be in all-caps.
2) in the results section, paragraph 2, the fourth line reads: "while the others had comorbid LD (n=70), MH disorders (n=22), or both MH (n=18)". I suspect the last point should read "both MH and LD (n=18)". This must be fixed or clarified.
3) in 3.1.4 Academic achievement, in the 3rd/4th line, the text says "one third of participants were underachievement". This must be fixed.
4) Results section – consistency is needed when referring to the different factors used in the hierarchical linear regression. On the table 1, the term “Cognitive variables” was used; in the written description of the results (section 3.1.4), these are referred to as ‘cognitive factors’. One term should be used for consistency of terminology. On table 1, the term “Behaviour Variables” is used. In the written description of the results (Section 3.1.4), in the reading section and the math section, these are referred to as “behavioural symptoms”; whereas in the spelling section, these are referred to as “behavioural factors”. Again, one term should be used to for consistency of terminology.
5) Table 1: I suggest considering adding to the description of Table 1 a legend of the different cognitive tests. Although the terms are defined in the body of the text, since many readers may be clinicians and not psychologists, it would be helpful to include the abbreviations and the full names ie WMI=Working Memory Index, etc
6) in Table 1, I suggest considering 'outdenting' the last two lines ie Total R2 and n. This would make it clearer that these are not part of Step 3/behavioural variables.
Secondly, While the sample characteristics are described in 3.1, a table may be helpful for this. I believe the sample characteristics are also relevant to the findings. In section 4.1, the authors discuss the large sample size contributing to the generalizability of the results. This is true. However, the sample characteristics require more examination and may be a limitation to the generalizability of the results.
Based on the numbers, ADHD diagnoses were made in n=184 children. ADHD only was n=74, or 40.2% of the diagnosed ADHD. That means that approximately 60% of the diagnosed children had comorbid conditions. This is a little on the lower side based on the literature, but it is in range.
The rates of comorbidity are a concern, and may limit generalizability. ADHD+LD is reported as n=70; and ADHD+LD+MH is n=18. This means that of all kids with ADHD, children with LD n=88. This yields a comorbidity rate of LD in ADHD in this sample of (88/184) 47.8%. This is on the higher end of the comorbidity of LD in ADHD based on the literature. Furthermore, when looking at children with a mental health (MH) diagnosis, they were reported as: ADHD +MH n=22; ADHD+MH+LD n=18 - this totals n=40 for children diagnosed with ADHD and a comorbid mental health disorder. Out of all children diagnosed with ADHD, this is 40/184 =21.7% who have ADHD with a comorbid MH diagnosis. This is a low rate compared to the published literature, where rates of comorbid anxiety diagnoses can be around 30%, oppositional defiant disorder - ranging from 30-50%, etc. (see Spencer TJ. ADHD and Comorbidity in Childhood. 2006 - J Clin Psychiatry - as one reference).
Since it appears that this clinic is a publicly funded clinic which provides a full psychological/educational assessment along with a clinical assessment, it is possible that the referrals skew toward children with a higher suspicion of learning difficulties/disabilities and a lower suspicion of MH concerns (where it may be possible that children with fewer learning concerns and more MH concerns may be referred to a publicly funded physician for ADHD assessment). Whether this is the case or not, it would be important for the authors to address this possible limitation. The results may not be as generalizable if the population is not representative of the typical childhood patient population referred for assessment for inattentive and hyperactive/impulsive symptoms.
Author Response
Please see the attachment.
Please see below for our responses. We have put our responses in red.
Reviewer #2 Comments:
This is excellent and relevant research. The authors have rigorously assessed and diagnosed children referred to their clinic; and all the data collected has allowed for an elegant statistical analysis. I note that the introduction is extremely well written - building the logic of the rationale for this research. The findings are very relevant to the field, as academic underachievement is the main reason for referral for ADHD assessment. The fact that not all of their subjects were diagnosed with ADHD allowed for a dimensional review of inattentive (INA) and hyperactive/impulsive (HI) behaviours, which is very helpful and relevant as well.
- Thank you for your positive review of this article.
Firstly, some smaller points:
1) in the title, ADHD should be in all-caps.
- This has been fixed.
2) in the results section, paragraph 2, the fourth line reads: "while the others had comorbid LD (n=70), MH disorders (n=22), or both MH (n=18)". I suspect the last point should read "both MH and LD (n=18)". This must be fixed or clarified.
- This has been fixed.
3) in 3.1.4 Academic achievement, in the 3rd/4th line, the text says "one third of participants were underachievement". This must be fixed.
- This has been changed to: “Overall, approximately one-third of participants were underachieving.”
4) Results section – consistency is needed when referring to the different factors used in the hierarchical linear regression. On the table 1, the term “Cognitive variables” was used; in the written description of the results (section 3.1.4), these are referred to as ‘cognitive factors’. One term should be used for consistency of terminology. On table 1, the term “Behaviour Variables” is used. In the written description of the results (Section 3.1.4), in the reading section and the math section, these are referred to as “behavioural symptoms”; whereas in the spelling section, these are referred to as “behavioural factors”. Again, one term should be used to for consistency of terminology.
- All mentions of cognitive factors have been removed and replaced with “cognitive variables”. All mentions of behavioural variables/factors have been removed and have been replaced with “behavioural symptoms”.
5) Table 1: I suggest considering adding to the description of Table 1 a legend of the different cognitive tests. Although the terms are defined in the body of the text, since many readers may be clinicians and not psychologists, it would be helpful to include the abbreviations and the full names ie WMI=Working Memory Index, etc
- We added a note under Table 3: “Note: VCI = Verbal Comprehension Index; VSI = Visual Spatial Index; WMI = Working Memory Index; PSI = Processing Speed Index; VMI = Visual-Motor Integration.”
6) in Table 1, I suggest considering 'outdenting' the last two lines ie Total R2 and n. This would make it clearer that these are not part of Step 3/behavioural variables.
- We created a space between Behavioural Symptoms (INA-Total and HI-Total) and Total R2 and n to make this clearer.
Secondly, While the sample characteristics are described in 3.1, a table may be helpful for this. I believe the sample characteristics are also relevant to the findings. In section 4.1, the authors discuss the large sample size contributing to the generalizability of the results. This is true. However, the sample characteristics require more examination and may be a limitation to the generalizability of the results.
- We added Table 1 (Demographic Characteristics of Total Sample) and Table 2 (Clinical Description of the Sample) to the results section.
Based on the numbers, ADHD diagnoses were made in n=184 children. ADHD only was n=74, or 40.2% of the diagnosed ADHD. That means that approximately 60% of the diagnosed children had comorbid conditions. This is a little on the lower side based on the literature, but it is in range.
- We agree. The slightly lower comorbidity rates are likely a result of the younger age of the sample.
The rates of comorbidity are a concern, and may limit generalizability. ADHD+LD is reported as n=70; and ADHD+LD+MH is n=18. This means that of all kids with ADHD, children with LD n=88. This yields a comorbidity rate of LD in ADHD in this sample of (88/184) 47.8%. This is on the higher end of the comorbidity of LD in ADHD based on the literature. Furthermore, when looking at children with a mental health (MH) diagnosis, they were reported as: ADHD +MH n=22; ADHD+MH+LD n=18 - this totals n=40 for children diagnosed with ADHD and a comorbid mental health disorder. Out of all children diagnosed with ADHD, this is 40/184 =21.7% who have ADHD with a comorbid MH diagnosis. This is a low rate compared to the published literature, where rates of comorbid anxiety diagnoses can be around 30%, oppositional defiant disorder - ranging from 30-50%, etc. (see Spencer TJ. ADHD and Comorbidity in Childhood. 2006 - J Clin Psychiatry - as one reference).
Since it appears that this clinic is a publicly funded clinic which provides a full psychological/educational assessment along with a clinical assessment, it is possible that the referrals skew toward children with a higher suspicion of learning difficulties/disabilities and a lower suspicion of MH concerns (where it may be possible that children with fewer learning concerns and more MH concerns may be referred to a publicly funded physician for ADHD assessment). Whether this is the case or not, it would be important for the authors to address this possible limitation. The results may not be as generalizable if the population is not representative of the typical childhood patient population referred for assessment for inattentive and hyperactive/impulsive symptoms.
- To address the above two points, we have added these sentences to the limitations section of the paper: “Additionally, the comorbidity of ADHD and LD in this sample was higher than average (47.8%), and the comorbidity of ADHD and MH was lower than average (21.7%), which limits the generalizability of these findings to the general population [83, 84]. This may be in part due to the younger age of this sample, and that the clinic was a partnership between mental health and the school board. This means that there was likely a referral bias toward children with learning problems rather than mental health problems.”
